# COVID-19: Immediate Predictors of Individual Resilience

**Regardt J. Ferreira [1,2,](#) , Fred Buttell [1,2] and Clare Cannon [2,3](#)**

[1] Tulane School of Social Work, Tulane University, New Orleans, LA 70112, USA; buttell@tulane.edu
[2] Department of Social Work, University of the Free State, Bloemfontein 9301, South Africa; cebcannon@ucdavis.edu
[3] Department of Human Ecology, University of California, Davis, CA 95616, USA
[*] Correspondence: rferrei@tulane.edu

**Abstract:** COVID-19 is a pandemic event not seen in a century. This research aims to determine important predictors of resilience towards the COVID 19/Coronavirus Pandemic. This study uses a cross-sectional design, with purposive snowball sampling, for primary survey data collected over 10 weeks starting the first week in April 2020. Participants completed a self-administered questionnaire on demographics and behavioral factors. Resilience was assessed using the 10-item Connor-Davidson Resilience Scale and perceived stress was assessed using the 10-item Perceived Stress Scale. 374 adults participated in the survey. OLS regression was performed to determine key associations among demographic variables, resilience measures, and perceived stress brought on by COVID-19. Age and education were statistically significantly positively associated with resilience, while English as a second language was significantly negatively associated. Participants who reported needing help from family and neighbors, total number of days in lockdown, and higher perceived stress were all significantly negatively associated with resilience. This study adds to immediate predictors of individual resilience to the ongoing infectious disease catastrophe created by the COVID-19 pandemic.

**Keywords:** resilience; social vulnerability; COVID-19; disasters; infectious disease

---

## 1. Introduction

The global outbreak of the novel coronavirus disease 2019 (COVID-19) has spread to every country across the world, infecting millions and killing hundreds of thousands of people [1], and warranting a pandemic declaration by the World Health Organization [2]. In its wake, COVID-19 has exacerbated mental health problems—such as anxiety and depression—the world over [3–7] created a global economic recession, not seen in modern history, and continues to threaten public health globally. COVID-19 has created a unique on-going disaster context, in which there are severe impacts on daily life including increased uncertainty, no end date, new fears related to contagion, illness, death, increases in a range of stressors, and reduced access to protective factors [8]. Moreover, many are fighting to recover from COVID-19 or are mourning lost loved ones to the disease without access to important cultural rituals. Important in the short and long term will be an individual's ability to maintain resilience in the face of COVID-19 and all its cascading and uncertain effects. The current study, building off of insights garnered from previous research into resilience in the wake of disease outbreaks and disaster, uses novel survey data collected during the first outbreak wave of COVID-19 to better understand important predictors of resilience to the current pandemic.



## 1.1. Resilience in the Face of Disaster

Resilience is understood here as an individual's ability to cope with risk, adversity, and stress, despite exposure to a serious stressor that may contribute to a variety of physical, behavioral, cognitive, and emotional symptoms [9–13]. Although the construct of resilience, its definition and scope, continues to be debated [11,14], important to resilience research is a primary focus on both protective and risk factors and how different stressors inform these factors [15–18]. Previous research into resilience and disasters suggests social and demographic variables impact resilience outcomes through uneven social vulnerabilities produced by systems of oppression. Social groups are differentially vulnerable to the harmful effects of disasters, which may impact their ability to cope and exhibit resilience [18–20].

Similarly, research indicates resilience has multiple dimensions that vary across age, gender, race/ethnicity, socioeconomic status, employment, English as a second language, and relationship status [18–24]. For instance, research suggests older individuals' resilience may be disproportionately impacted by their common health problems [9,25,26], while studies have shown disasters exacerbate existing mental and physical health problems [27]. Moreover, research has found that race contributes to social vulnerability through the lack of access to resources and the social, economic, and political marginalization due to racial disparities [19,28–30]. Research has also found that gender is another robust predictor of social vulnerability to natural hazards [31] and that low educational attainment increases social vulnerability [32].

Research suggests that socially vulnerable populations are also more likely to perceive greater stress in the wake of disaster [16,33]. Perceived stress is the degree to which one perceives the threat of a stressor and how well one can behaviorally and cognitively adapt to it [34,35]. Previous research, using the Perceived Stress Scale (PSS), found that Hispanic ethnicity, overall exposure to disaster (i.e., Hurricane Sandy), and reported mental health issues were positively associated with increased perceived stress [33]. Moreover, research suggests that perceived stress is negatively correlated with resilience [36]. Resilience, in turn, may also affect a person's perception of stress, which can have implications for one's quality of life and health [37,38].

Although most people demonstrate resilience in the face of highly aversive events, research has shown that there tend to be lower rates of resilience to infectious disease outbreaks [10]. Research also suggests that, over time, those affected by aversive events can overcome distress and become resilient [39]. For instance, in their study of the SARS outbreak in 2003, Bonanno and colleagues found that approximately half of SARS survivors recovered from distress and continued to be resilient [10,39]. In more recent cross-sectional nationwide studies of Chinese people during the COVID-19 pandemic found that although many displayed depressive and anxiety symptoms, middle-aged respondents, who took precautionary measures and engaged with accurate COVID-health information showed decreased psychological distress [4,5].

Lastly, potentially traumatic events (PTEs) can be exacerbated by experiences of disaster, COVID-19, [22,40,41]. Although most people are exposed to at least one PTE in their lifetime, only a small number of people exhibit enough loss or trauma responses to meet the criteria threshold for posttraumatic stress disorder [22]. Most people seem to recover from aversive effects and can successfully navigate PTEs [10]. Previous research into PTEs suggests that resilience outcomes may be higher during or immediately following a PTE and that a delayed moderate or severe trauma effect, such as grief, may not occur for one to two years post-event [10,40]. Because of the unique context, COVID-19 creates—prolonged and intermittent lockdowns, social isolation, a global recession—it is necessary to better understand the risk and protective factors that may contribute to resilience immediately. Understanding these factors in the COVID-19 context will enable us to better advise policymakers, regulators, and public health officials on how to support and advance resilience and health in affected populations during the current infectious disease crisis.



## *1.2. Purpose of the Study*

The overall purpose of this study is to determine the important predictors of resilience towards the COVID 19/Coronavirus Pandemic. The guiding research question is to investigate how much of the variance in individual resilience can be explained by demographic variables and personal experiences related to the COVID-19 pandemic. More specifically, the purpose of this study is three-fold and aims to (1) investigate the role of perceived stress, current situation and demographic variables on overall resilience during the COVID-19 outbreak; (2) present findings from a study active during the COVID-19 pandemic; (3) add to the scant literature on disasters, infectious disease, and resilience.

## 2. Methods

The current study uses a cross-sectional design. Data used with this study was collected over a 10-week period from an online survey launched the first week of April 2020. The Tulane University Social/Behavioral Institutional Review Board approved the study. The online survey was distributed through one of the author's social media accounts (e.g., Facebook, Instagram, and LinkedIn) and advertised on the Tulane School of Social Work social media outlets and website for a period of 10 weeks. Inclusion criteria for the online survey required participants older than 18 years only and who had access to the survey link. Exclusion criteria included those who were younger than 18 years. The survey focus was on (a) their previous disaster experience, (b) their resilience (i.e., the Connor Davidson Resilience Scale), (c) perceived stress (i.e., Perceived Stress Scale), (d) current situation as it relates to COVID-19, and (e) personal and household demographics. The online Qualtrics survey took an estimated 10 min to complete.

## *2.1. Participants*

The study sample consisted of individuals having access to the online Qualtrics survey link. Participants were recruited for participation in the study through a mixture of snow-ball sampling with one study author requesting participants to share the survey link on their social media accounts, as well as having the survey link displayed on the School of Social Work's home page and in media outlets for the school. The sample for this study includes 374 adults who completed the online survey. SPSS 26 was utilized to conduct the final data analysis.

## *2.2. Measures*

### 2.2.1. Outcome Variable

Resilience is the outcome variable used for this study. The 10-item Connor Davidson Resilience Scale (CD-RISC 10) was administered to study participants as the measure of resilience. This study used the CD-RISC 10, which is a well-established, abbreviated version of the original 25-item CD-RISC. The scale utilizes a 5-point Likert scale ranging from 1 for "not at all" to 5 for "nearly all the time [21]. The CD-RISC 10 has displayed high internal consistency, construct validity, and test-retest reliability [21,42,43]. The 10-item scale has proven to have strong psychometric properties in general, as well as across various demographic indicators, including gender, age, and race/ethnicity [44–47]. The CD-RISC 10 asks respondents to rate their own resilience by responding to the following ten statements: (1) I am able to adapt when changes occur; (2) I can deal with whatever comes my way; (3) I try to see the humorous side of things when I am faced with problems; (4) Having to cope with stress can make me stronger; (5) I tend to bounce back after illness, injury, or other hardships; (6) I believe I can achieve my goals, even if there are obstacles; (7) Under pressure, I stay focused and think clearly; (8) I am not easily discouraged by failure; (9) I think of myself as a strong person when dealing with life's challenges and difficulties; and, (10) I am able to handle unpleasant or painful feelings like sadness, fear, and anger. Scores on the CD-RISC 10 range from 0 to 40 and most studies use 32 as the cut-off score for resilience.

2.2.2. Predictor Variables

Demographic Variables

Theoretically, the study is guided by variables associated with Cutter et al.'s [19] Social Vulnerability Index and Cannon et al.'s [20] work on social vulnerability. This theoretical framing was used to identify and select the relevant demographic variables that would be included in this survey, which was being distributed within the context of a disaster situation, the COVID-19 pandemic. The foundation of the approach is that those who are under stress or who have experienced a traumatic event, such as a disaster, are more prone to be socially vulnerable [16–18]. The extant literature and this theoretical framing led us to include the following demographic variables and response categories: (1) age; (2) gender (1 = male, 2 = female); (3) race (1 = white, 2 = minority); (4) relationship (1 = not in a relationship, 2 = in a relationship) (5) employment (1 = employed, 2 = unemployed); (6) education (1 = less than high school, 2 = high school, 3 = some college, 4 = associate degree, 5 = bachelor's degree, 6 = graduate degree), (7) speak other language than English at home (1 = yes, 0 = no); (8) residential status (1 = own house, 2 = rent, 3 = other).

Measure of Perceived Stress

The study also included a recoded variable for a total score on the Perceived Stress Scale (PSS). The PSS show correlations with a variety of health-related measures that includes stress measures, self-reported health and health services measures, health behavior measures, smoking status and health-seeking behaviors [48]. Higher values on the PSS indicate greater perceived stress. The PSS was categorized into low (1), medium (2), and high (3).

COVID-19 Experiences

The purpose of the following set of predictor variables is to determine which experiences associated with COVID-19 influenced individual resilience. The following predictor variables and response categories were included: (1) has the COVID-19/Coronavirus lead to you losing your job (1 = yes; 0 = no); (2); has the COVID-19/Coronavirus lead to you losing income (1 = yes; 0 = no); (3) do you think you will need help and cooperation from others (e.g., family, friends, or neighbors) to recover from the impact of COVID-19/Coronavirus? (1 = very little help, 2 = little help, 3 = neither very little help nor very much help, 4 = much help, 5 = very much help); (4) do you think you will need help and cooperation from others (e.g., government or non-governmental organizations) to recover from the impact of COVID-19/Coronavirus? (1 = very little help, 2 = little help, 3 = neither very little help nor very much help, 4 = much help, 5 = very much help); (5) how many days have you spent in "stay at home/lockdown".

## 3. Results

The sample of 374 participants had a mean age 47.01 (SD = 14.67), 74.6% females (*n* = 279) and 25.4% men (*n* = 95). The majority of the sample identified as White 86.1% (*n* = 322). In terms or relationship status, the majority of participants were in a relationship, 55.9% (*n* = 209). The majority of the sample were employed with 63.4% (*n* = 237) reporting being employed at the time of study participation. Regarding education, only 1.3% of respondents had less than a high school diploma (*n* = 5), followed by 3.5% (*n* = 13) some college, 12.8% (*n* = 48), associate's degree 4.5% (*n* = 17), bachelor's degree 25.7% (*n* = 96) and the majority of the sample had a graduate degree 52.1% (*n* = 195). The majority of the sample spoke English at home 71.1% (*n* = 268). Nearly two-thirds of the sample reported owning their home with 63.9% (*n* = 239). The mean number of days spent in "stay at home/lockdown" for the sample is 28.3 days (SD = 18.12).

According to the perceived stress scale measure, 33.5% (*n* = 123) were categorized as having low stress, compared to 60.5% (*n* = 222) having medium stress and 6% (*n* = 22) reporting high stress. For the

outcome variable, the CD-RISC 10, respondents had a mean score of 30.97 (SD = 5.46) for the 10-item scale. Table 1 provides a detailed description of the demographic variables for these two groups.

**Table 1.** Demographic characteristics of the sample.

| Characteristic | Participants (*n* = 374) | | |
|---|---|---|---|
| | Mean/% | *n* Range | SD |
| **Age (in years)** | 47.01 | 369 20–83 | 14.66 |
| **Gender** | | | |
| Male | 25.4 | 95 | |
| Female | 74.6 | 279 | |
| **Race** | | | |
| White | 86.1 | 322 | |
| Minority | 13.9 | 52 | |
| **Relationship** | | | |
| In a relationship | 55.9 | 209 | |
| Single | 44.1 | 165 | |
| **Employment** | | | |
| Employed | 63.4 | 137 | |
| Unemployed | 36.6 | 237 | |
| **Education** | | | |
| Less than 12 years/No HS Diploma | 1.3 | 5 | |
| HS Diploma/GED | 3.5 | 13 | |
| Some College | 12.8 | 48 | |
| Associate Degree | 4.5 | 17 | |
| Bachelor Degree | 25.7 | 96 | |
| Graduate Degree | 52.1 | 195 | |
| **English Second Language** | | | |
| English | 71.7 | 268 | |
| English Second Language | 28.3 | 106 | |
| **Residential** | | | |
| Own House | 63.9 | 239 | |
| Rent | 36.1 | 135 | |
| **Time—Stay at Home/Lockdown** | | | |
| Number of days at home | 28.3 days | | 18.12 |
| **CD-RISC 10 Mean Score** | 30.97 | 364 9–40 | 5.46 |
| **Perceived Stress Scale** | | | |
| Low | 33.5 | 123 | |
| Medium | 59.4 | 222 | |
| High | 6.0 | 22 | |
| **Perceived Stress Mean Score** | 16.37 | | 6.27 |

*3.1. Resilience Model Testing*

To answer the research question regarding how much of the variance in individual resilience among survey respondents can be explained by demographic variables and experiences related to COVID-19, two separate sets of regression models were performed. The first multiple linear regression

model investigated demographic variables as predictors of resilience, using the CD-RISC 10 as the dependent variable (see Table 2). The second multiple linear regression model focused on experiences during the COVID-19 pandemic as predictors of resilience and also used the CD-RISC 10 as the dependent variable (see Table 3). Significant coefficients for each model are identified below and reported in Tables 2 and 3, respectively. All assumptions of the models were met.

**Table 2.** An ordinary least square (OLS) multiple linear regression for demographic predictors of resilience.

| | B | β | *t* | *p* | 95% CI Lower Bound | 95% CI Upper Bound |
|---|---|---|---|---|---|---|
| Age | 0.1 ** | 0.26 | 4.56 | 0.001 | 0.06 | 0.14 |
| Gender | −1.29 | −0.10 | −1.95 | 0.052 | −2.6 | 0.01 |
| Race | 0.29 | 0.02 | 0.36 | 0.317 | −1.3 | 1.86 |
| Relationship | −0.32 | −0.03 | −0.52 | 0.600 | −1.52 | 0.88 |
| Employment | 0.59 | 0.05 | 0.96 | 0.340 | −0.63 | 1.81 |
| Education | 1.02 ** | 0.24 | 4.45 | 0.001 | 0.57 | 1.48 |
| English Second Language | −1.48 * | −0.12 | −2.29 | 0.023 | −2.75 | −0.21 |
| Residential | 0.59 | 0.05 | 0.9 | 0.366 | −0.7 | 1.89 |

Notes: *n* = 358; df = 8; * *p* < 0.05; ** *p* < 0.001.

**Table 3.** OLS multiple linear regression for COVID-19 predictors of resilience.

| | B | β | *t* | *p* | 95% CI Lower Bound | 95% CI Upper Bound |
|---|---|---|---|---|---|---|
| COVID−19/Coronavirus lead to you losing your job | −1.1 | −0.08 | −1.43 | 0.155 | −2.61 | 0.42 |
| COVID−19/Coronavirus lead to you losing income | −0.49 | −0.05 | −0.84 | 0.402 | −1.65 | 0.66 |
| Cooperation from others (e.g., family, friends, neighbors) | −0.69 * | −0.17 | −2.71 | 0.007 | −1.18 | −0.19 |
| Cooperation from others (e.g., government or non−governmental organizations) | 0.11 | 0.03 | 0.45 | 0.655 | −0.36 | 0.57 |
| Total days spent in "stay at home/lockdown" | −0.03 * | −0.11 | −2.11 | 0.036 | −0.06 | −0.002 |
| Perceived Stress Scale Category | −3.52 ** | −0.39 | −7.2 | 0.001 | −4.48 | −2.56 |

Notes: *n* = 291; df = 6; * *p* < 0.05; ** *p* < 0.001.

### 3.1.1. Social Demographics

An Ordinary Least Square (OLS) multiple linear regression analysis was performed to determine if demographic variables (i.e., age; gender; race; relationship; employment; education; speak other language then English at home; and, residential status) were related to an increased level of self-reported resilience among the sample. The $R^2$ statistic was statistically significant $F_{(8,358)} = 6,092$, $p = 0.0001$, $R^2$ adjusted = 0.120, indicating that 12.0% of the variance in resilience can be explained by demographic predictor variables. A summary of the regression coefficients is presented in Table 2. Results indicate that as age and educational attainment increase, there are corresponding increases in resilience, a relationship which was statistically significant. The dummy coded variable for English as second language was

statistically significant and negatively associated with resilience, indicating that respondents for whom English is a second language fair worse on the resilience outcome compared to those who only speak English at home.

### 3.1.2. COVID-19 Experiences

An OLS multiple linear regression analysis was performed to investigate if COVID-19 experiences—measured by the following survey questions: has the COVID-19/Coronavirus led to you losing your job; has the COVID-19/Coronavirus led to you losing income; do you think you will need help and cooperation from others (e.g., family, friends, or neighbors) to recover from the impact of COVID-19/Coronavirus; do you think you will need help and cooperation from others (e.g., government or non-governmental organizations) to recover from the impact of COVID-19/Coronavirus; how many days have you spent in "stay at home/lockdown"; perceived stress scale category (1 = low, 2 = medium, 3 = high)—were related to an increased level of self-reported resilience among the sample. The $R^2$ statistic was statistically significant $F$ (6,291) = 11.941, $p$ = 0.001, $R^2$ adjusted = 0.198 indicating that 19.8% of the variance in resilience can be explained by COVID-19 experiences predictors. A summary of the regression coefficients is presented in Table 3. Results indicate that as respondents reported needing greater help and cooperation from others (e.g., family, friends, or neighbors) to recover from the impact of COVID-19/Coronavirus, their resilience decreased, a statistically significant relationship. The more days spent under "stay at home orders/lockdown", the less resilient participants reported, a relationship that was also statistically significant. Finally, higher total scores on the perceived stress scale corresponded to a statistically significant decrease in resilience. Perceived stress was the most robust predictor of resilience in this analysis.

## 4. Discussion

The results of this study are important because the study took place amid an infectious disease pandemic alongside what would soon become crippling national job losses resulting in widespread financial hardship. Although this sample was overwhelmingly female (75%), well educated (i.e., 26% had an undergraduate degree and 52% had a graduate degree) and employed (63% were employed), the total sample score on the CD-RISC 10 was a 30.97 (SD = 5.46), indicating that the sample was on the cusp of being considered resilient by that instrument. This is important for several reasons. First, this study occurred towards the beginning of the CODID-19 pandemic. Very few studies investigating the relationship between disaster and resilience have been able to be conducted while the disaster is taking place. Our results suggest that at just 28 days, on average, into the COVID-19 pandemic, 66% of this sample were reporting moderate or high levels of stress. It begs the question of whether populations with less social capital and fewer financial resources would be reporting even higher levels of stress and lower levels of resilience. Second, this snapshot of the sample from the beginning of the pandemic is concerning in its own regard, but things have rapidly deteriorated since the time of the study. In the U.S., case counts and death tolls are increasing at this time and many states are rolling back many of the openings that came with shifting into Phase 3, which would mean opening up higher-risk workplaces for instance, and are returning to Phase 2, which entails limiting time spent outside one's home and limiting travel to only permissible activities (i.e., health care, food, exercise). It seems quite likely that the people that comprised this sample are presently more stressed and less resilient now than they were at the time of survey participation in April and May, as uncertainty and confusion abound. Finally, COVID-19 is a global pandemic with no end in sight, which distinguishes it from other forms of natural (e.g., hurricanes) and man-made disasters. As an infectious disease with no cure or vaccine on the horizon, it seems likely that the second administration of this survey would find more profound levels of stress and decreased levels of self-reported resilience even among the well-educated and employed.

The regression model investigating the relationship between demographic variables and self-reported resilience found that both age and education successfully indicated greater levels

of resilience, while English as a second language reduced it. This finding of a positive relationship among age, education, and resilience is not surprising as there is ample empirical evidence in the resilience literature that establishes this connection [16,17]. However, given the unique nature of COVID-19 as an infectious disease that disproportionately targets the old, this finding is a little surprising in this context. Clearly, the fact that the average age of the sample was 47 at least partially explains this positive relationship for this group but it is possible that as the virus continues to spread in the U.S. this relationship may also be changing. Specifically, even if the study was repeated with a sample of participants roughly the same age as this sample, the new information that is constantly emerging about how the virus spreads and precautions people should take to prevent the unintentional spread to older family members suggests that this positive relationship might disappear. The model also indicated an inverse relationship between resilience and speaking English as a second language. This finding is also predictable given the vast economic hardships caused by the pandemic. It is easy to see how navigating both governmental and health care bureaucracies are made more difficult by not speaking English as a first language [29].

The regression model investigating the relationship between COVID-19 experiences and self-reported resilience found that as respondents need to rely more on others, had to stay home longer under the lock-down and reported higher levels of stress on the PSS, resilience dropped. This finding seems to provide evidence for the contention earlier that the time period for the study might have minimized some of the longer-term effects of the pandemic-induced lockdown. The average time of the lockdown for this sample was 28 days and at the time of writing, we are approaching 120 days. Perhaps equally important, however, is that the mood in the U.S. has also shifted more towards pessimism during the interim. During the time period for this study, contemporary wisdom was that the lockdowns would be temporary and would allow hospitals to not be overrun with COVID-19 cases. However, it seems that there is no hard end in sight for social distancing and mask-wearing, which likely would exacerbate the stress reported by participants in a second wave of the study. An interesting finding in this model is that resilience went down as participants reported they had to rely on others for help. This question takes on an additional meaning in the face of the COVID-19 pandemic because, as an infectious disease, respondents may have feared for the health and safety of themselves and their loved ones as they had to interact with others for aid and support.

*Limitations and Future Directions for Research*

One obvious limitation is the nature of the sample of respondents. They were predominantly female, white, educated, and employed. Not mentioned in the survey but also a limitation is that they were computer literate and had access to wi-fi, which suggests they were more affluent than would be typical of a broad community sample of participants. Although important for quickly gaining a better understanding of resilience, infectious disease, and disasters in a constantly evolving context, this research uses a cross-sectional snapshot of factors related to resilience. Future research should consider employing longitudinal analyses to better understand the long-term impacts on resilience.

## 5. Conclusions

This study adds to the scant literature on resilience within the context of a pandemic. We aimed to determine the predictors of resilience in the face of an infectious disease catastrophe. Given the findings from the study, governments must mitigate the associated risks of a pandemic by providing the needed resources for individuals, households, and communities to maintain resilience over a long period of time. The uncertain end of COVID-19 requires governments to offer a buffer against the pandemic impact and to ultimately reduce stress to create optimal health and well-being for citizens facing adversity.

**Author Contributions:** Conceptualization, R.J.F. and F.B.; methodology, R.J.F. and F.B.; software, R.J.F.; validation, R.J.F., C.C. and F.B.; formal analysis, R.J.F.; investigation, R.J.F.; resources, R.J.F., C.C., F.B.; data curation, R.J.F.; writing—original draft preparation, R.J.F., C.C., F.B.; writing—review and editing, R.J.F., C.C., F.B.; supervision,

R.J.F. and F.B.; project administration, R.J.F. All authors have read and agreed to the published version of the manuscript.

**Funding:** This research was funded by the Carol Lavin Bernick Faculty Grants at Tulane University.

**Conflicts of Interest:** The authors declare no conflict of interest.

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
