# Peer review of "COVID-19: Immediate Predictors of Individual Resilience"

_sustainability, doi:10.3390/su12166495_

Round 1

Reviewer 1 Report

The manuscript is well written.

This study aimed to predict the factors associated with resilience during this COVID situation. I feel this is an important study at this time. 

I have a few suggestions to the authors. The authors mentioned inline 64 and 65 that there will be an exacerbation of pre-existing mental and physical problems. In this situation, I was wondering whey the preexisting mental problems were not included in the characters and demographic predictors. Including their pre-existing mental illness may have a significant impact on the outcome with regression analysis(1). 

And also I was wondering whether the participants were also a vulnerable group like front line health care workers or social care staff(1). The methods section mentioned that the survey was distributed through the authors' personal social media accounts. So there might be high chances that the participants might be health care/social workers. If there is a concern, it should be mentioned in the limitations.    

The rationale for a positive relationship between age and resilience was well explained. 

The authors acknowledged the limitations of involving technically sophisticated people in the study. 

(1) DOI:https://doi.org/10.1016/S2215-0366(20)30168-1

Reviewer 2 Report

SARS-CoV pandemic affected deeply our societies in all aspects. The biggest burden it posed was the phycological one on people.

In this context the paper by Ferreira et al covers individual resilience and its predictors, offering a new insight on how the pandemic affected everyday life.

The paper is well written and the research well designed. Apart from some typos it should be considered for publication in present form.

Reviewer 3 Report

Thank you for submitting this interesting article. Let me positively highlight the idea to study the Immediate Predictors of Individual Resilience of COVID-19. Due to the complexity of the material of this article, as well as my desire (and obligation) to contribute to the overall quality of your article with my review, I have invested a lot of effort to review your work. Effort and time have been invested in studying some of your written references, as well as in general consideration of the broader subject and the background "scene" of the subject of your work, the COVID-19 pandemic.

In general, I consider this article of yours to be a valuable scientific contribution to the overall studies of the situation and environment associated with the (still ongoing) COVID-19 pandemic. But please, let me introduce some general remarks and objections.

Introduction

The part of the ‘Introduction’ section that describes the broad pandemic environment and this study wide ‘framework’, as well as, a development of the idea of this study is well and briefly written. My suggestion is to change used reference no. 3 (Wan, W., 2020) with some scientific findings about exacerbated mental health problems in the world because it could contribute to the overall scientific value of your study. Wan is expert journalist of Washington Post and he used some ‘scientific’ findings in his article (eg., nearly one-half of Americans responding to a Kaiser Family Foundation poll say their mental well-being has suffered; experts say …; the psychological trauma ranges from anxiety and depression to substance abuse, post-traumatic stress disorder, and suicide). Later in your study you have provided some of that such scientific literature (like Qiu et al., 2020 or Wang et al., 2020), and I know that there is a lot of recently published studies about mental health problems during COVID-19 pandemic.

The part of the ‘resilience’ and the ‘purpose’ section into the Introduction is also very well written.

Methods

The methods part of the study is very concisely and well written.

I suggest you to consider to make some changes concerning the description of Perceived Stress Scale (PSS) measure. Either that part of text should be transferred to ‘COVID-19 Experiences’ section or it could be it positioned as ‘independent’ section in the ‘predictor variables’ section. Placing of PSS description in the ‘Demographic Variables’ paragraph is not appropriate. As well, in section ‘COVID-19 Experiences’ in lines 200-201 three used measures of PSS Category are defined as one of the experiences predictor variables.

Results

The results part of the study is well written and the results are presented in tables in a very good manner. OLS multiple regression analysis is an appropriate statistical procedure for collected data.

Table 1 is placed into the middle of one sentence (line 212-213), so consider to move it below. Also, table 1 should have a tittle next to the number of table.

Tables 2 and 3 should have a tittle next to the number of table (line 251; 276) and placed outside the table.

Style of presenting the results should be equally, especially regarding the names of variables and regarding of using bolding style. It is not appropriate to bold only ‘significant’ results because you already used the signs for that purpose (* or ***).

My opinion is that it is appropriate for any result to present it with just two digits next to point, except when it is used for p-value result (three digits) for the purpose of determining of the exact significance level. As well, in Table 1. you have used just one digit for all demographic variables, and two digits for the means and SDs, respectively. Although SPSS is providing always three digits results in their tables, I think that is redundant for social sciences. It makes reading and understanding of the tables much more difficult.  

Discussion

Explanation and interpretation of the results were made in a reasonable and in a very understandable manner.

In line 298, Phase 3 and Phase 2 in US were mentioned. Please, explain that phases briefly to the international readers which do not know the regulations for COVID-19 pandemic in US!

Conclusion

Conclusion is very short, good and reasonable.

References

All the references should be changed from APA style to CHICAGO style as proposed by the journal (https://www.mdpi.com/journal/sustainability/instructions), and by the MDPI publisher (https://www.mdpi.com/authors/references). Athough I have found some differences in referencing regulations proposed in these two sources and attached manuals (Reference List and Citations Style Guide for MDPI Journals; Reference List and Citations Style Guide, For MDPI Humanities and Social Sciences Journals), that changes should be made.

Specific comments:

  1. Line 95, please, decide what to use, either using reference in the parentheses (preferred by the journal instructions) or using reference in text with names of authors (APA style).
  2. Line 98, decide, reference in text or reference in parentheses.
  3. Lines 155-156, decide, reference in text or reference in parentheses.
  4. Line 251, please, check underlying of P value into the table legend (notes). As well, consider all my previous comments about tables.
  5. Line 299, explain briefly what it means, Phase 2 and Phase 3 in the US.
  6. Lines 408-410 and 460-461, present the same reference no. [12] and [30], authors Ferreira, R.J., Adolph, V., Hall, M., & Buttell F. (2019). Predictors of individual resilience... Please, correct it here, and in the text lines.
  7. Line 491, finish (merge) previous reference with that text.
